# Exploring online sensor parameters as proxies for polar organic chemicals—An innovative approach for combined sewer overflow monitoring

Laura Waldner[1,2]*, Viviane Furrer[1], Pierre Lechevallier[1,2], Fabienne Maire[1,2], Heinz Singer[1], Lena Mutzner[1]*

1 Department of Urban Water Management, Swiss Federal Institute of Aquatic Science and Technology (Eawag), Dübendorf, Switzerland, 2 Institute of Environmental Engineering (IfU), ETH Zürich, Zurich, Switzerland

* laura.waldner@eawag.ch (LW); lena.mutzner@eawag.ch (LM)

## Abstract

Combined sewer overflows (CSOs) can release toxic organic chemicals into surface waters during rain events. Currently, most overflow sites are not monitored because commonly used methods, such as automated grab sampling followed by laboratory analysis using liquid chromatography coupled with mass spectroscopy (LC-MS), are costly and time-consuming. Due to this monitoring gap, the dynamics of organic chemicals in CSOs remain poorly understood. This study explores the use of eight online sensor parameters as proxies for polar organic chemicals from different sources in combined sewer systems during wet weather. We used sensor and organic chemical data collected in three urban catchments of varying sizes. Correlations between chemicals from the same source and sensor parameters were calculated. In the largest catchment (160,000 inhabitants), indoor chemicals are strongly correlated with flow, electrical conductivity, spectral absorption coefficient at 254 nm ($SAC_{254\,nm}$), and ammonium ($NH_4$-N). Additionally, linear regressions were developed to predict organic chemical concentrations from sensor data. Models based on $SAC_{254\,nm}$ and $NH_4$-N predict indoor chemical concentrations with median relative errors of 32% and 29%, respectively, in the large catchment. Prediction performance for road chemicals is independent of catchment size, with median relative errors ranging from 39% to 44%, using either level or flow measurements. However, the prediction of pesticide concentrations remains limited, as these chemicals exhibit diverse patterns across rain events. Overall, our results suggest that linear regression models can estimate indoor chemical concentrations in large catchments and road chemical concentrations in catchments of any size. However, for real-world implementation, further research is needed to refine calibration requirements and validate the models across diverse catchments. Nevertheless, these models are

**Data availability statement:** All files are available through Eawag's Research Data Institutional Collection (ERIC/open) at https://doi.org/10.25678/000ETY.

**Funding:** LM received a SNSF Ambizione grant [grant no. PZPGP2_209133], Swiss National Science Foundation, https://www.snf.ch/en/N18L3oGWomTSSGkF/funding/careers/ambizione. VF received a grant from the Swiss Federal Office for the Environment (FOEN) [grant no. 19.0071.PJ / 8971CB0BA], Swiss Federal Office for the Environment, https://www.bafu.admin.ch/bafu/en/home.html. PL was supported by the EU H2020 research and innovation program [grant no. 101008626] (Co-UDLabs Project), European Commission, https://research-and-innovation.ec.europa.eu/funding/funding-opportunities/funding-programmes-and-open-calls/horizon-2020_en. The funders had no role in study design, data collection and analysis, decision to publish, or preparation of the manuscript.

**Competing interests:** The authors have declared that no competing interests exist.

promising for cost-effective, long-term monitoring of organic chemicals and for mitigating the impact of CSO discharges.

## 1. Introduction

When combined sewer systems exceed their capacity during rain events, they can discharge toxic organic chemicals into surface waters through combined sewer overflows (CSOs). These discharges of a mixture of wastewater and stormwater contain numerous organic chemicals [1,2] that threaten aquatic species [3–5]. Wastewater contains chemicals from domestic and industrial activities such as food additives, pharmaceuticals and personal care products (PPCPs), and corrosion inhibitors [6]. Urban runoff, also termed stormwater, contains chemicals washed off urban surfaces such as biocides, plant protection products (PPPs), and tire wear leachates [1,2,7–11]. In addition, some chemicals occur in similar concentrations in both wastewater and stormwater, such as the industrial chemicals di(2-ethlyhexyl)phthalate, 4-nonylphenol, and the herbicide carbamazepine [1,12].

Monitoring organic chemicals in CSOs enables the early detection and quantification of harmful substances while assessing the effectiveness of control measures and treatment strategies. Such monitoring is crucial to mitigating emissions and reducing adverse impacts on aquatic ecosystems. However, organic chemical concentrations exhibit high spatiotemporal variability. Particularly, organic chemicals from municipal wastewater, also termed indoor chemicals, exhibit very high concentration fluctuations in small catchments due to pulsed wastewater flushes from households [13,14]. Therefore, data collection at high temporal resolution and at a large number of discharge sites is required to capture their dynamics [2,13–16]. A sampling interval of two minutes is recommended to capture the actual dynamics of organic chemicals in CSOs [13,14]. Particularly, organic chemicals from municipal wastewater, also termed indoor chemicals, exhibit very high concentration fluctuations in small catchments due to pulsed wastewater flushes from households [13,14]. State-of-the-art monitoring is performed by automated grab sampling followed by laboratory analysis using liquid chromatography coupled with mass spectroscopy (LC-MS) [8–11,13]. However, this approach is unsuitable for continuous monitoring because it is cost- and time-intensive, requires automated sampling, and provides only discrete time points [14,17–19]. Hence, innovative, cost-effective monitoring approaches are needed to monitor organic chemicals in sewers during wet weather [20].

Online water quality sensors in urban drainage systems offer the potential to provide affordable and continuous monitoring of a range of water quality parameters, including water level, pH, turbidity, UV-Vis absorption, ammonium, nitrate, potassium, chloride, phosphorus, and chemical oxygen demand [21–26]. The cost and maintenance requirements vary by sensor type, with advanced sensors such as UV-Vis spectrophotometers or ion-selective electrodes requiring higher investment costs and maintenance efforts (S1 Table 8 in S1 File). However, sensor-based monitoring generally remains significantly less costly and time-intensive than traditional approaches such as automated grab sampling combined with

LC-MS analysis, especially in the long term (S1 Tables 8 and 9 in S1 File). Moreover, sensors enable continuous, high-resolution data collection, which is crucial for capturing contaminant dynamics in real-time. However, current online sensors do not directly measure organic chemicals and thus must rely on proxy parameters. For example, in wastewater treatment plants (WWTPs), the spectral absorption coefficient at 254 nm ($SAC_{254\,nm}$) is used as a proxy for micropollutant removal [27–29]. Launay et al. [30] and Carpenter et al. [31] used different environmental covariates as proxies for organic chemicals in surface waters. However, little is known about the potential of online sensor parameters as proxies for polar organic chemicals in urban drainage systems. Because some organic chemicals exhibit distinct temporal patterns in sewer systems depending on their origin [32], they might be captured through sensor proxies that reflect the same temporal dynamics.

This study investigates the potential of online sensor parameters to serve as proxies for polar organic chemicals in combined sewers during wet weather using measurements from three catchments of various sizes. We aim to answer two questions: (i) Can sensor parameters serve as proxies for polar organic chemicals in sewers? (ii) Can sensor-based models predict the concentrations of polar organic chemicals in sewers?

Using sensor proxies, we were able to predict concentrations of polar indoor chemicals and chemicals from road runoff with median relative errors of about 30% and 40%, respectively. However, we were unable to identify a suitable sensor proxy for many polar organic chemicals from outdoor sources, such as PPPs and biocides, that exhibit highly variable concentration dynamics. With further refinements, the developed models could serve as a cost-effective tool for long-term monitoring of indoor and road runoff chemicals in sewer systems and for mitigating the impacts of CSO discharges on surface waters.

## 2. Materials and methods

### 2.1. Study sites

Data for this study were collected by Furrer et al. [13,32] and Lechevallier et al. [33] and published as openly accessible datasets in the Eawag Research Data Institutional Collection (ERIC/open) [34,35]. As the data were already published and the authors conducted no new sampling, this study did not require any permits. The data were collected in three urban catchments in Switzerland that ranged in size from 2,700 to 159,000 inhabitants (catchments S, M, and L, Table 1) [13,32,33]. In catchments S and L, combined sewage was sampled at CSOs to analyze polar organic chemicals [13,32]. In catchment M, wastewater from a combined sewer bypass was sampled from a flow that was temporarily diverted from the main sewer to facilitate sampling and measurement [33]. Further information on the catchments and the sampling can be found in Furrer et al. [13,32] for catchments S and L and in Lechevallier et al. [33] for catchment M.

Table 1. Characteristics of the three catchments investigated.

| Characteristic | Catchment S | Catchment M | Catchment L |
|---|---|---|---|
| Population equivalents (PE) | n.a. | 24,800 | 184,000 |
| No. of inhabitants [-] | 2,700 | n.a. | 159,000 |
| Impervious area [ha$_{red}$] | 17 | 71 | 368 |
| Land use | Rural village, little industry, agriculture | Peri-urban residential area, major outskirt shopping and industrial area, agriculture, leisure-use airport | Several villages and towns, agriculture, a small amount of industry |
| No. of rain events analyzed (Total no. of samples) | 19 (299) | 4 (86) | 13 (270) |

n.a. indicates that a value is not available.

## 2.2. Sampling during rain events

**Start of sampling.** This study defined a rain event period during which the flow exceeds a certain threshold. During such an event, a series of samples was taken. S1 Table 1 in S1 File provides an overview of the rainfall characteristics of the events, the sampling times, and the flow rates during the sampling period. In catchment S, combined sewage was sampled during 19 rain events by Furrer et al. [13,32]. The sampling of a rain event was initiated when the flow in the sewer exceeded the system capacity of 288 $m^3$/h, indicating a CSO discharge. Sampling continued until the flow fell below 288 $m^3$/h or until all bottles of the autosampler were full. In catchment M, four rain events were sampled by Lechevallier et al. [33]. Sampling started when the flow in the sewer increased above its dry weather baseline (300–380 $m^3$/h, depending on the time of day), calculated from average dry weather flows. Sampling ended when it fell below the dry weather baseline or when all bottles of the autosampler were full. For rain event M.2, sampling started 50 minutes late due to problems with the autosampler. Despite the missing initial phase, rain event M.2 was retained for analysis due to the limited size of the dataset (comprising only four rain events). The potential impact of the delayed sampling start is discussed in section 3.4. Limitations. In catchment L, 13 rain events were sampled. Sampling for all rain events except two was initiated when the flow in the sewer exceeded the system capacity of 5,400 $m^3$/h [13,32]. However, for rain events L.11 and L.12, sampling originally included a one-hour dry weather period before the onset of rainfall [32]. Furrer et al. [32] sampled pre-rainfall to investigate whether pre-event dilution dynamics would enhance correlations between indoor-sourced chemicals and additional parameters (e.g., $NH_4$-N, dissolved organic carbon). However, as no significant improvement in correlation was found [32], we excluded these pre-rainfall samples from our analysis to ensure consistency across all events. The flow thresholds defined reflect differences in sewer system capacity and catchment characteristics, such as the number of connected people and the size of the connected impervious area.

**Sampling procedure.** Samples for analysis of polar organic chemicals were collected either as grab samples or as time-weighted composite samples in 1 L glass bottles using automated samplers (TP5 C, MAXX Mess- und Probenahmetechnik GmbH). Grab samples (250 mL) were taken every 10 minutes. For composite sampling, subsamples were taken every 2 minutes, 150 mL in catchments S and L, and 50 mL in catchment M. Five subsamples were combined to a composite sample (750 mL or 250 mL) representing 10 minutes. S1 Table 2 in S1 File provides an overview of the sample types and the number of samples collected during each rain event. Further details on the sampling procedure can be found in Furrer et al. [13,32] and in Lechevallier et al. [33].

## 2.3. Polar organic chemicals

**Selection of chemicals.** Twenty substances were selected to investigate the dynamics of polar organic chemicals from different sources in the three catchments investigated. The selection criteria included polarity, high occurrence and concentration, representation of various urban sources, and expected source-specific behavior. Among the selected chemicals, triclosan, 1,3-diphenylguanidine, diuron, and mecoprop have been identified as major contributors to ecotoxicological risk in CSO discharges [3]. The selected compounds are included in both datasets reported by Furrer et al. [35] and Lechevallier et al. [33]. Only Carbendazim data is missing for catchment L. Following Furrer et al.'s [32] non-target screening results, chemicals from different urban source clusters were selected: indoor chemicals from a cluster of over 3,000 substances with similar dynamics, and road runoff chemicals from a cluster of over 150 substances. Polar substances were selected because they are understudied in the context of CSOs. Following Zessner et al. [36], a threshold of $LogK_{ow} \leq 5$ was chosen for polarity to ensure dissolution and transport in the water phase. Table 2 shows the target analytes, along with their assigned groups, categorized by their urban sources. Compounds were assigned to the categories "indoor", "road", and "plant protection products & biocides" based on a literature review of their typical uses. Substances with multiple contributing sources were categorized as "diverse" to reflect their mixed origins. More information on the use of these chemicals, their $LogK_{ow}$ values, and LC-MS analysis is presented in S1 Tables 3–6 in S1 File.

**Table 2. Groupings of selected polar organic chemicals.**

| Chemical group (source) | Polar organic chemicals (abbreviation) |
|---|---|
| Indoor (municipal wastewater) | Acesulfame (ACE), caffeine (CAF), cyclamate (CYC), candesartan (CAN), citalopram (CIT), diclofenac (DCF), hydrochlorothiazide (HCT), triclosan (TCS) |
| Road (stormwater) | 1,3-diphenylguanidine (DPG), 6PPD-quinone (DPG), hexa(methoxymethyl)melamine (HMMM) |
| Plant protection products (PPPs) & biocides (stormwater) | 2,4-dichlorophenoxyacetic acid (2,4-D), carbendazim (CBZ), diuron (DCMU), 1-methyl-4-chlorophenoxyacetic acid (MCPA), mecoprop-p (MPP), 2-n-octyl-4-isothiazolin-3-on (OIT) |
| Diverse (municipal wastewater & stormwater) | 1H-benzotriazole (BT), 4-&5-methylbenzotriazole (MeBT), N,N-diethyl-meta-toluamide (DEET) |

Polar organic chemicals were grouped according to the urban source/use.

**Chemical analysis.** After sampling, the water samples were stored at −20°C in muffled glass vials. After defrosting at room temperature and centrifugation, the target analytes were identified by liquid chromatography coupled with high-resolution tandem mass spectrometry (LC-HRMS/MS). The samples were injected directly into the LC-HRMS/MS system, using both positive and negative modes of electrospray ionization. Samples from catchments S and L were measured with a Q-Exactive orbitrap mass spectrometer (Thermo Fischer Scientific Inc.), and samples from catchment M were measured with a 6495C triple quadrupole LC-MS system (Agilent Technologies Inc.) [13,32,33]. Target analytes were quantified using reference standards and isotope-labelled internal standards. Furrer et al. [13,32] and Lechevallier et al. [33] provide further details on the analytical methods. Organic chemical data are openly available from Furrer et al. [35] (catchments S and L) and Lechevallier et al. [34] (catchment M).

## 2.4. Online sensors

During the rain events studied (S1 Table 1 in S1 File), the combined sewage characteristics were measured with a range of online sensors in parallel with automated water sampling. Various sensor measurements were taken in the three catchments (Table 3). Although $NH_4$-N can theoretically be measured with an ion-selective electrode sensor, this study used laboratory values because no sensor data are available for catchments S and L, and sensor measurements in catchment M proved unreliable [33]. S1 Table 7 in S1 File provides an overview of the sensor models, measurement intervals, and locations, as well as the maintenance and calibration procedures for all sensors used in this study. Further details on sensor measurements can be found in Furrer et al. [13,32] and in Lechevallier et al. [33]. Sensor data is available from the ERIC/open repositories of Furrer et al. [35] (catchments S and L) and Lechevallier et al. [33] (catchment M).

## 2.5. Data analysis

**Programming language.** Python (version 3.9.18 [37]) was used for data preprocessing and analysis, due to its widespread use in scientific research and its robust libraries for data processing and modeling.

**Table 3. Sensor data available at the measurement sites in catchments S, M, and L.**

| | Rain | Sewer level | Sewer flow | Temperature | EC | pH | Turbidity | $SAC_{254\,nm}$ | $NH_4$-N |
|---|---|---|---|---|---|---|---|---|---|
| Catchment S | ✓ | ✓ | ✓ | X | ✓ | X | X | X | ✓(lab) |
| Catchment M | ✓ | ✓ | ✓ | ✓ | ✓ | ✓ | ✓ | ✓ | ✓(lab) |
| Catchment L | ✓ | X | ✓ | ✓ | ✓ | ✓ | X | ✓ | ✓(lab) |

Abbreviations: EC: electrical conductivity, $NH_4$-N: ammonium, $SAC_{254\,nm}$: spectral absorption coefficient at 254 nm.

**Pretreatment of sensor data.** Lechevallier et al. [33] describe the data preprocessing procedures applied to sensor data from catchment M in detail. These included timestamp harmonization, outlier removal based on sensor-specific validity ranges, and quantitative quality checks. In contrast, Furrer et al. [13,32] did not apply specific preprocessing steps to the sensor data from catchments S and L, except for timestamp harmonization. Therefore, this study identified and replaced single outliers using linear interpolation (see S1 section 5. Details on data analysis in S1 File). No additional filtering or smoothing was applied to data from catchments S and L to avoid introducing subjective bias.

Sensor data were preprocessed to match the sampling time points. For grab samples, only sensor values at the exact sampling time were retained; for composite samples, the mean of sensor values measured at the five subsample time points was calculated. If sensor readings did not align with the (sub-)sampling times, sensor values were linearly interpolated.

**Linear correlations.** Pearson correlation coefficients (PCCs) were calculated to assess the relationships between organic chemicals within a chemical group and between organic chemicals and sensor parameters (see S1 section 5. Details on data analysis in S1 File). Correlations with PCC > 0.5 and $p$-value < 0.05 were considered indicators of similar temporal dynamics. Such correlations suggest either source-specific behavior (section 3.1. Source-specific behavior of polar organic chemicals in CSOs) or that a sensor can serve as a proxy for a chemical group (section 3.2. Correlations between polar organic chemicals and sensor parameters).

Correlations between organic chemicals within the same chemical group were calculated using non-normalized concentrations (section 3.1. Source-specific behavior of polar organic chemicals in CSOs). To calculate PCCs between a chemical group and a sensor parameter (section 3.2. Correlations between polar organic chemicals and sensor parameters), both chemical concentrations and sensor values were normalized using the Z-score (see S1 section 5. Details on data analysis for equation in S1 File). The normalized concentrations of all chemicals within a chemical group were combined into a single dataset, which was then used to determine correlations with sensor data.

**Linear regression models.** Simple linear regression (SLR) models and multiple linear regression (MLR) models were developed to investigate whether a sensor parameter (SLR) or a combination of sensor parameters (MLR) can predict measured organic chemical concentrations. Models were fitted to absolute organic chemical concentrations using the ordinary least squares linear regression function. Because this function cannot handle missing values, time points when a chemical was not detected (concentration below the limit of detection) were removed from the dataset. Missing sensor values were replaced by linear interpolation unless two or more consecutive values were missing, in which case either the sensor or the rain event was excluded from modelling, depending on the size of the remaining dataset, as detailed in S1 section 5. Details on data analysis in S1 File.

To identify which sensor best predicted each chemical group, an SLR model was fitted to the normalized concentrations (Z-score) of all chemicals in each chemical group (Table 3) for every available sensor (Table 2). For models based on water level and flow, the dry weather baseline was also included as a predictor to account for stormwater dilution. For indoor chemicals, level and flow models also incorporated average time-shifted level and flow to account for the observed time lag in dilution. S1 section 6. Analysis of dilution time lag of organic chemicals from municipal wastewater explains the calculation and results of the time shift analysis. For every chemical group, the best sensor parameter was the one that fitted the model with the highest $R^2$.

Further, MLR models were fitted to the normalized concentrations (Z-score) of all chemicals of each chemical group to determine whether a combination of sensor parameters improves predictions. For each chemical group, a stepwise regression function using bidirectional elimination was used to identify the most relevant sensor parameters for prediction. In each step, predictors with $p$-value < 0.01 were retained, and predictors with $p$-value > 0.05 were removed.

Model performance across rain events was assessed with cross-validation. The dataset was divided into sets for model fitting (all rain events except one) and sets for validation (event excluded from the fitting set). For each chemical, models were trained on the fitting sets with the best predictor(s) for the chemical group and then applied to the validation set for

prediction. The model fits and predictive performance were evaluated by calculating the $R^2$ between concentration measurements and fits and the absolute and relative root mean square error (RMSE and RRMSE) between measurements and predictions. $R^2 \geq 0.5$ and RRMSE $\leq 50\%$ were considered indicators of good model fit and prediction, respectively (S1 section 5. Details on data analysis in S1 File).

## 3. Results and discussion

### 3.1. Source-specific behavior of polar organic chemicals in CSOs

We analyzed three catchments of different sizes (S, M, L) to investigate whether organic chemicals show source-specific behavior depending on catchment size.

**Indoor chemicals show source-specific dilution patterns in large catchments.** Furrer et al. [32] found that indoor chemicals exhibit source-specific dilution behavior in catchment L but not in catchment S. Adding catchment M to the data supports their finding that source specificity becomes more distinct as catchment size increases (Fig 1a). The correlation (median PCC) rises from 0.33 to 0.70 to 0.89 in catchments S, M, and L (Fig 1a) as dilution patterns become more pronounced in larger catchments (S1 Figs 3–5 in S1 File). In small catchments, pulsed wastewater flushes from households can cause large concentration fluctuations, but more numerous point sources and longer transport distances disperse this effect in larger catchments [13,14]. However, because only three catchments are studied here, the minimum catchment size required for a clear source-specific dilution pattern remains uncertain. Based on our data, this threshold appears to lie between 2,000 and 20,000 inhabitants, but it likely depends on the frequency of a chemical's use. The observed relationship between catchment size and dilution pattern of indoor chemicals aligns with the mechanistic understanding that larger catchments smooth out individual pulses due to a large number of pulses and dispersion effects, suggesting that this trend is generalizable to other urban catchments.

**Chemicals from road runoff show source-specific patterns independent of catchment size.** Organic chemicals from road runoff show source-specific behavior in catchments S and L, as indicated by the strong correlations (median PCC $> 0.5$, Fig 1b) [32]. In catchment M, however, only 1,3-diphenylguanidine and 6PPD-quinone correlate strongly (PCC $= 0.79$, S2 Table 5 in S2 Tables), and correlations with HMMM are weak (PCC $= 0.07$ and 0.11, S2 Table 5 in S2 Tables). HMMM is more rapidly mobilized during rain events M.2 and M.4 (S1 Figs 4 and 7 in S1 File), lowering the median PCC to 0.11 (Fig 1b). Other studies have also reported first flushes of tire-wear-derived contaminants in streams, suggesting transport-limited wash-off [38–40]. An additional clustering analysis grouped all road runoff chemicals in

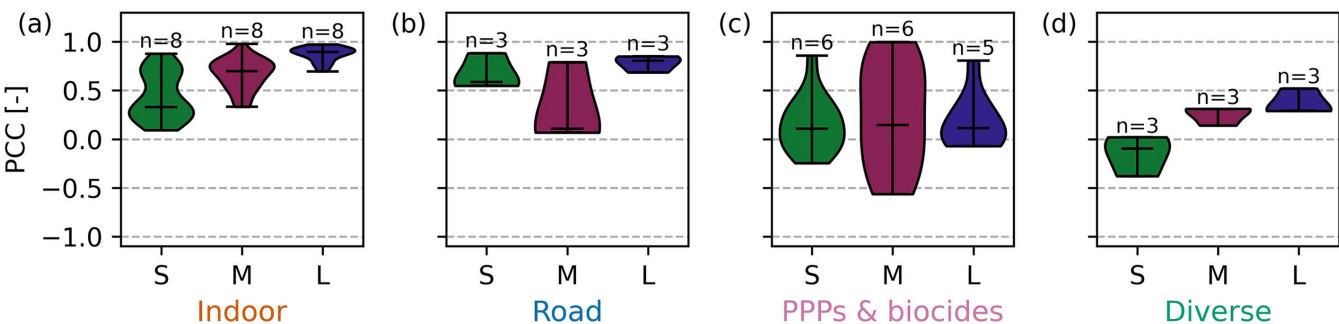

**Fig 1. PCCs for correlations between polar organic chemicals of one chemical group.** Fig 1 shows the PCCs for correlations between polar organic chemicals within each chemical group (indoor (a), road (b), PPPs & biocides (c), and diverse (d)) across all rain events for the catchments S, M, and L. The bar indicates the median and *n* is the number of substances in each catchment and group. Further statistical measures of the correlations (*p*-value, confidence interval, etc.) can be found in S2 Tables 4–6 in S2 Tables. Abbreviations: PCC: Pearson correlation coefficient, PPPs: plant protection products.

every catchment (S1 Table 11, S1 Figs 3–5 in S1 File). This confirms that road runoff chemicals exhibit distinct dynamics regardless of catchment size, despite the variations observed in HMMM mobilization.

**PPPs and biocides show fluctuating concentrations.** PPP and biocide concentrations show no source-specific behavior (median PCC < 0.2, Fig 1c) or first flush effects, except for diuron and OIT during rain event M.3 (S1 Fig 7 in S1 File, [32]). PPP and biocide concentrations fluctuate widely across all catchments. This aligns with previous studies that have reported diverse temporal patterns for these groups of stormwater pollutants [32,41]. Biocides can either diffuse continuously from building materials or show first-flush emissions due to surface accumulation during dry periods. Pesticide concentrations in urban runoff can peak with rainfall or remain constant in sewer discharges [13,41]. Hence, fluctuating concentrations can result from the mobilization of various sources at varying times during rain events, combined with the complexity of drainage systems and transport distances [41,42].

**Chemicals from diverse sources show dilution patterns in the large catchment.** 4-&5-methylbenzotriazole, benzotriazole, and DEET correlate more strongly in the larger catchments, similarly to indoor chemicals. The median PCC increases from −0.10 in catchment S to 0.31 in catchment M and then to 0.29 in catchment L (Fig 1d). Our supplementary clustering analysis also indicates that these chemicals exhibit a dilution pattern similar to those of indoor chemicals in catchment L (S1 Table 11, S1 Fig 5 in S1 File). We conclude that 4-&5-methylbenzotriazole, benzotriazole, and DEET primarily originate from households in the larger, more urbanized catchments M and L, whereas their source remains unclear in the small catchment S.

### 3.2. Correlations between polar organic chemicals and sensor parameters

Linear correlations between chemical groups and sensor parameters were calculated for each catchment to investigate whether sensor measurements can serve as proxies for polar organic chemicals.

**EC, SAC$_{254\,nm}$, and NH$_4$-N are suitable proxies for indoor chemicals in large catchments.** Indoor chemicals correlate negatively with sewer flow and level and positively with electrical conductivity (EC), the spectral absorption coefficient at 254 nm (SAC$_{254\,nm}$), and ammonium (NH$_4$-N) (Fig 2a). Negative correlations of indoor chemicals with flow and level result from the dilution of municipal wastewater by stormwater. EC, organic carbon (indicated by SAC$_{254\,nm}$), and ammonium are typically higher in wastewater than in stormwater [1,43,44], hence the correlation observed with indoor chemicals. Correlations with NH$_4$-N are the strongest correlations in all catchments (highest PCC, Fig 2a), likely because ammonium originates from urine, the primary excretion pathway of many indoor chemicals. However, correlations with EC or SAC$_{254\,nm}$ are similarly strong in catchments M and L, and correlations with flow in catchment L. Thus, NH$_4$-N is a key proxy, but other sensors may also serve as indicators depending on catchment size.

Correlations between indoor chemicals and sensor parameters are stronger in larger catchments (Fig 2a), where dilution patterns are more pronounced (S1 Figs 3–5 in S1 File). All indoor chemicals show clear dilution patterns and correlate strongly with EC, SAC$_{254\,nm}$, and NH$_4$-N in catchment L (PCC > 0.7, S1 Figs 5 and 11 in S1 File). In contrast, chemicals that are not frequently used show fluctuating concentration patterns and thus weak correlations in catchments S and M (PCC < 0.5, S1 Figs 3, 4, and 11 in S1 File). This suggests that for indoor chemicals, the performance of sensor proxies increases with increasing catchment size.

We also analyzed correlations between indoor chemicals and additional parameters. In catchment M, the UV-Vis spectrum is available from 228 to 708 nm. We find that most indoor chemicals correlate better with light absorption at wavelengths around 230 nm than at 254 nm (S1 Table 13 in S1 File), where human urine and anionic surfactants absorb light [45]. Furthermore, we observe that the maximal dilution of indoor chemicals often occurs after the flow peak in all three catchments (S1 Fig 1 in S1 File). Possible explanations for this phenomenon include the resuspension of sewer deposits, which release organic chemicals [1,46], or stormwater parcels being transported faster than wastewater parcels (S1 section 6. Analysis of dilution time lag of organic chemicals from municipal wastewater in S1 File). Analyzing correlations with time-shifted flow and level reveals stronger correlations for some chemicals, particularly in catchment M (S1 Fig 13

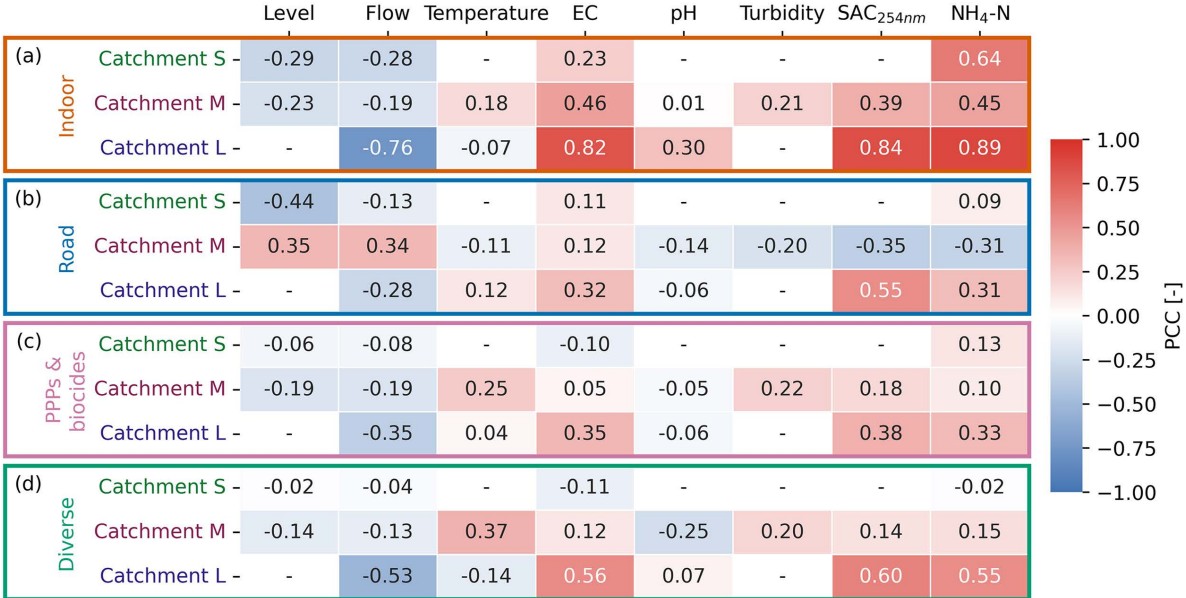

**Fig 2. PCCs for correlations between polar organic chemicals of each chemical group and sensor parameter.** Fig 2 shows the PCCs for correlations between polar organic chemicals of each chemical group (indoor (a), road (b), PPPs & biocides (c) and diverse (d)) and sensor parameters across all rain events in the catchments S, M, and L. Positive PCCs are colored red and negative PCCs are colored blue. "-" indicates that a correlation coefficient can not be calculated because insufficient data is available. Further statistical measures of the correlations (*p*-value, confidence interval, etc.) can be found in S2 Tables 7–9 in S2 Tables. Abbreviations: EC: electrical conductivity, NH4-N: ammonium, PCC: Pearson correlation coefficient, PPPs: plant protection products, $SAC_{254\,nm}$: spectral absorption coefficient at 254 nm.

in S1 File), where the time shift shows the smallest standard deviation and thus is most consistent across rain events (S1 Table 12 in S1 File).

**Level and flow are not universally valid proxies for chemicals from road runoff.** No strong correlations are observed between the road runoff group and any sensor in any catchment (PCC < 0.5, Fig 2b), except for $SAC_{254\,nm}$ in catchment L. We expected positive correlations with flow and level due to rapid surface runoff from impervious surfaces [38] and negative correlations with $SAC_{254\,nm}$ and $NH_4$-N due to the dilution of wastewater by stormwater [1,46]. These correlations are weak in catchment M (Fig 2b). In contrast, in catchments S and L, the road runoff group shows weak negative correlations with flow and level and weak to moderate positive correlations with $SAC_{254\,nm}$ and $NH_4$-N (Fig 2b). However, examination of correlations from individual rain events reveals that the single chemicals often correlate positively with level and flow and negatively with $SAC_{254\,nm}$ and $NH_4$-N in catchments S and L (S1 Figs 15 and 17 in S1 File). The fact that these correlations are not reflected when data from all rain events is combined suggests that factors other than stormwater flow influence the dynamics of organic chemicals from road runoff. These factors may include the preceding dry period, source location, rain intensity, and traffic volume [32,47–49]. Although road runoff chemicals show source-specific patterns in every catchment (S1 Table 11 in S1 File), no sensor parameter is uniquely related. Thus, a universally valid sensor proxy has yet to be found.

**High variability of PPPs and biocides limits the identification of sensor proxies.** PPPs and biocides show weak correlations with all sensor parameters in every catchment studied (PCC < 0.5, Fig 2c). No sensor parameter proxy can be identified for this chemical group because their diverse, fluctuating concentrations are not reflected by any sensor parameter available (S1 Fig 11 in S1 File). Given these chemicals' lack of source-specific behavior [32], finding a sensor proxy for this chemical group may remain infeasible. For example, Carpenter et al. [31] developed separate models with different predictors for every pesticide studied in surface water.

**Proxies for indoor chemicals are also applicable to other wastewater-derived chemicals.** In catchments S and M, chemicals from diverse sources correlate weakly with all sensor parameters (PCC < 0.5, Fig 2d). In catchment L, however, 4-&5-methylbenzotriazole, benzotriazole, and DEET correlate strongly with flow, EC, $SAC_{254\,nm}$, and $NH_4$-N (PCC > 0.5, Fig 2d, S1 Fig 11 in S1 File), as they exhibit similar source-specific dilution patterns as the indoor chemicals (S1 Fig 5 in S1 File). Thus, EC, $SAC_{254\,nm}$, and $NH_4$-N can also serve as proxies for 4-&5-methylbenzotriazole, benzotriazole, and DEET when they primarily originate from indoor sources. However, no reliable sensor proxy has been found for situations where both wastewater and stormwater are significant sources of these chemicals.

### 3.3. Predicting polar organic chemical concentrations from sensor parameters

We used two types of models, single linear regression (SLR) and multiple linear regression (MLR), to identify sensor parameters that can predict chemical groups (Table 3). We investigated first whether single sensors can predict absolute organic chemical concentrations (SLR models) and second, whether predictions improve when more sensor parameters are added (MLR models).

**Models based on $NH_4$-N, $SAC_{254\,nm}$, or EC predict indoor chemical concentrations accurately in large catchments.** The predictive performance of the SLR models depends on catchment size, organic chemical, and sensor parameter. In catchments S and L, $NH_4$-N is the sensor parameter that fits the SLR model with the highest $R^2$ for all indoor chemicals, whereas in catchment M, level provides the best fit (Table 4). Predictions using these sensors (Table 4) yield median relative errors (RRMSE) of 64%, 79%, and 29% in catchments S, M, and L, respectively (Fig 3c). Hence, the model's predictive accuracy improves with increasing catchment size (Fig 3c, S1 Fig 19 in S1 File) as dilution patterns of indoor chemicals become more stable (S1 Figs 3–5 in S1 File). In the small catchment S, the model fits, and therefore, the predictions are only good for selected indoor chemicals (acesulfame, caffeine, cyclamate, citalopram, and hydrochlorothiazide) ($R^2 > 0.5$, S3 Table 15 in S1 File) that show a dilution pattern and correlate well with sensor proxies (S1 Figs 3 and 11 in S1 File). In catchment M, model fits are generally poor (median $R^2 = 0.37$, Fig 3a, S1 Fig 19 in S1 File) as correlations with level are weak (PCC = −0.23, Fig 2a). Additionally, overfitting to the training set is an issue (S3 Table 2 in S3 Tables) due to the small size of the dataset, making model results in catchment M uncertain. Overall, our

**Table 4. Sensors selected to predict each chemical group for the catchments S, M, and L.**

| Chemical Group | Catchment | Level | Dry level | Shifted level | Flow | Dry flow | Shifted flow | Temperature | EC | pH | Turbidity | $SAC_{254\,nm}$ | $NH_4$-N |
|---|---|---|---|---|---|---|---|---|---|---|---|---|---|
| Indoor | S | • | • | | | | | | • | | | | X• |
| | M | X | X | X• | | | | • | | • | • | • | • |
| | L | | | | | • | • | • | • | | | | X• |
| Road | S | X• | X | | | • | | | | | | | • |
| | M | X• | X• | | | | | | | | | | • |
| | L | | | | X | X• | | • | | | | • | • |
| PPPs & biocides | S | | • | | X• | X• | | | • | | | | |
| | M | | | | | | | X• | | | | | |
| | L | | | | | | | | | • | | X• | |
| Diverse | S | | | | • | | | | • | | | | X• |
| | M | | • | | | • | | X | | • | | • | |
| | L | | | | | | | • | | • | • | X• | |

× indicates the sensor that fits the SLR model with the highest $R^2$ for all chemicals of a chemical group (indoor, road, PPPs & biocides, and diverse). • indicates the sensors selected by stepwise regression for the MLR model fitted to all chemicals of a chemical group. Abbreviations: EC: electrical conductivity, $NH_4$-N: ammonium, PPPs: plant protection products, $SAC_{254\,nm}$: spectral absorption coefficient at 254 nm.

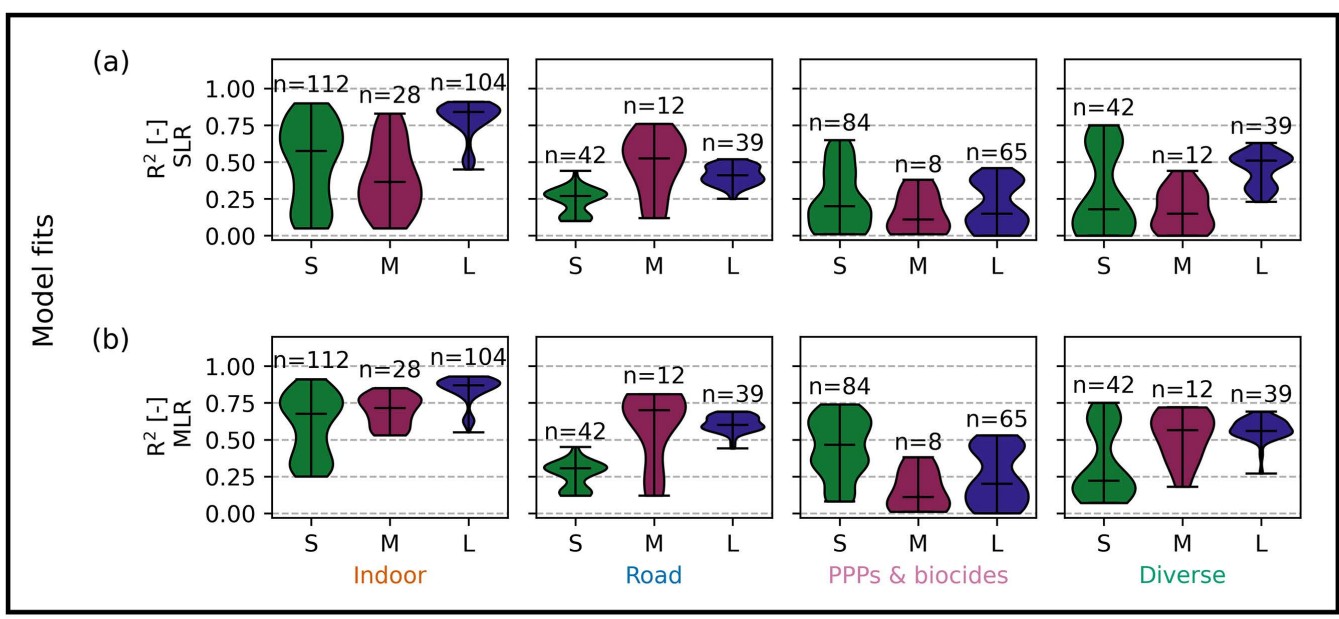

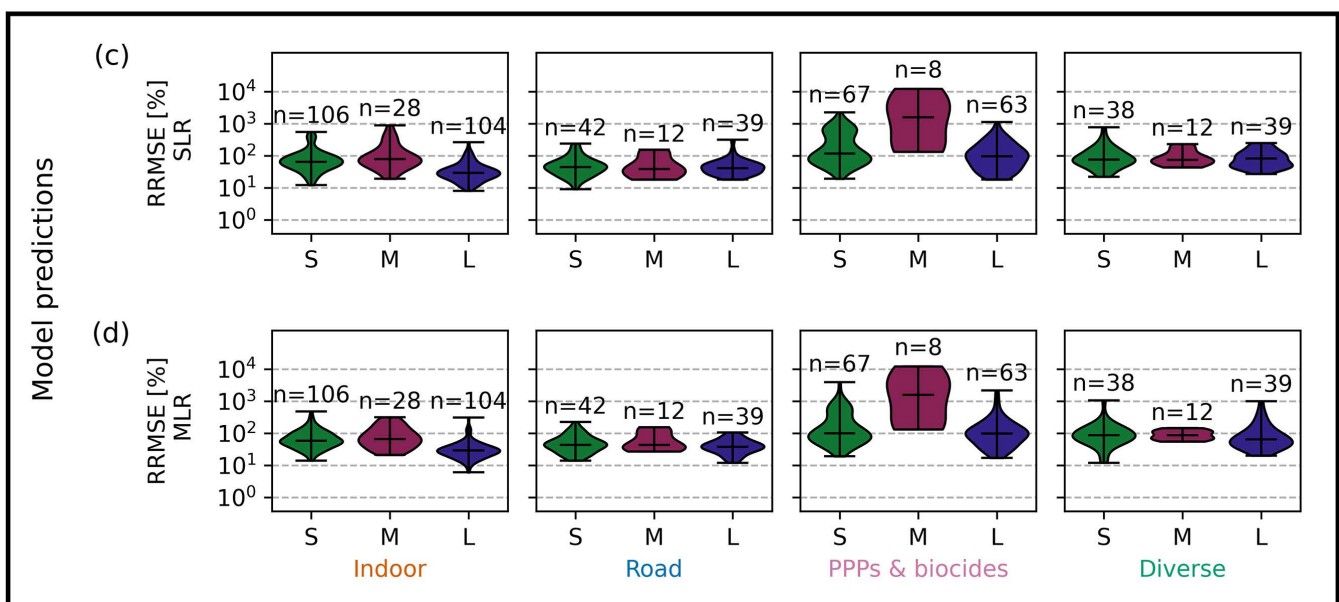

**Fig 3. Comparison of SLR and MLR model fits and predictions for organic chemical groups.** The violin plots in the upper panel show the $R^2$ of the SLR models **(a)** and MLR models **(b)** fitted to data from all rain events except one (training set). The violin plots in the lower panel show the RRMSE of the SLR models **(c)** and MLR models **(d)** predicting data from the excluded event (testing set). For every chemical, a model was fitted with the sensors shown in Table 4. These models then predicted the chemical concentrations in the excluded event. The results of all chemicals of a chemical group (indoor, road, PPPs & biocides, and diverse) are agglomerated into one violin plot. *n* indicates the number of model fits (upper panel) and the number of model predictions (lower panel). The horizontal line indicates the median. S3 Tables 1–20 in S3 Tables show further statistical measures of the models displayed.

results suggest that SLR models can effectively predict indoor chemical dynamics in large catchments where chemicals exhibit source-specific dilution patterns.

For indoor chemicals, sensors other than the sensor fitting the SLR model with the highest $R^2$ (Table 4) also perform well. In catchment M, predictions using $SAC_{254\,nm}$ and $NH_4$-N yield median relative errors of 70% and 76% and thus are better than predictions using level (S1 Fig 28 in S1 File). The better prediction performance of models using sensor parameters other than level further suggests that overfitting is an issue in catchment M. In catchment L, SLR models using EC and $SAC_{254\,nm}$ yield median relative errors of 33% and 32% (S1 Fig 28 in S1 File). Hence, these models perform similarly to models using $NH_4$-N, but offer the practical advantage of easier sensor measurement.

When fitting MLR models for indoor chemical concentrations, additional sensors (level, flow, temperature, EC, pH, turbidity, $SAC_{254\,nm}$, and/or $NH_4$-N) are identified as predictors (Table 4). Adding predictors improves the model fit from 0.58 to 0.68, from 0.37 to 0.72, and from 0.84 to 0.87 in catchments S, M, and L, respectively (Fig 3b). However, the predictive performance of the MLR models is not much better than that of the SLR models. Adding more predictors reduces the RRMSE by 6% and 13% in catchments S and M, but does not improve the prediction in catchment L (Fig 3c and 3d). These larger improvements in model fit compared to predictions indicate overfitting; the added predictors capture noise rather than the actual signal. Thus, using one sensor parameter ($NH_4$-N or $SAC_{254\,nm}$) is sufficient for predicting indoor chemicals in large catchments, but the relative error remains relatively large in smaller catchments.

**Models based on flow or level approximately predict concentrations of road runoff chemicals.** The performance of SLR models for road runoff chemicals is independent of catchment size. Level or flow provided the best model fits for road chemicals in all catchments (Table 4). Models using these parameters show fits with median $R^2$ values of 0.27, 0.53, and 0.41, and median relative prediction errors of 44%, 39%, and 41% for catchments S, M, and L, respectively (Fig 3a and 3c). The errors are higher when chemical concentrations fluctuate independently of flow (S1 Figs 3–5, 21, 23, and 25 in S1 File), as is the case for HMMM during two rain events in catchment M (S1 Fig 4 in S1 File). Hence, model fits are poorer for HMMM (median $R^2 = 0.22$, S3 Table 2 in S3 Tables) than for 1,3-diphenylguanidine and 6PPD-quinone in catchment M (median $R^2 = 0.64$ and 0.51, S3 Table 2 in S3 Tables). Additionally, flow and level alone cannot explain prolonged elevated concentrations during some rain events (S1 Figs 3–5 in S1 File). Consistent with section 3.2. Correlations between polar organic chemicals and sensor parameters, models using level or flow can only partially predict the dynamics of road runoff chemicals.

When MLR models are fitted to road chemicals, parameters quantifying the proportion of wastewater are added as predictors in addition to level or flow. $NH_4$-N is added in all catchments, and $SAC_{254\,nm}$ and temperature are included in catchment L (Table 4). As with indoor chemicals, these additional predictors improve the model fit but not the prediction accuracy, likely due to overfitting (Fig 3). The median RRMSE decreases by 1% and 3% in catchments S and L, respectively, and increases by 4% in catchment M (Fig 3c and 3d). This suggests that $NH_4$-N and $SAC_{254\,nm}$ cannot explain a significant proportion of road chemical dynamics, and level or flow alone are sufficient predictors. Both SLR and MLR models exhibit broad residual distributions, indicating limited predictive accuracy with the current set of sensor parameters (S1 Figs 19 and 20 in S1 File). Hence, additional proxies need to be identified to account for other influencing factors. Current studies report mixed findings on the influence of other environmental factors, such as antecedent dry period, rainfall intensity, and traffic volume and speed, on chemicals from road runoff [32,47–49]. This underscores the need for further research to clarify the transport and leaching mechanisms of these substances and to better understand how these factors are interrelated.

**Linear models based on sensor proxies cannot predict PPP and biocide concentrations.** SLR models fail to predict PPP and biocide concentrations. Flow, temperature, and $SAC_{254\,nm}$ are selected as predictors in catchments S, M, and L, respectively (Table 4). However, models based on these sensors result in poor model fits (median $R^2 = 0.20$, 0.11, and 0.15 in catchments S, M, and L, Fig 3a), as they do not capture the PPP and biocide dynamics (S1 Fig 19 in S1 File). Accordingly, they yield RRMSEs of 116%, 1,588%, and 96% with broad ranges from minimum 19% to maximum 2,248%,

from 132% to 12,214%, and from 18% to 1,112%, in catchments S, M, and L, respectively (Fig 3c). The very high prediction errors in catchment M likely result from limited data (mecoprop and diuron for four rain events). Adding more predictors does not improve predictions significantly (Fig 3c and 3d). The poor model performance can be attributed to the lack of source-specific behavior of this chemical group and the fact that the used sensor proxies do not accurately reflect the strongly fluctuating and variable concentration patterns of any of the PPPs and biocides studied (S1 Figs 11, 19, and 20 in S1 File). Linear models, based on linear combinations of sensor inputs, cannot capture such irregular temporal dynamics. Thus, the sensor-based models developed in this study are unable to predict PPP and biocide concentrations accurately.

**Linear models can predict concentrations of chemicals from diverse sources when their primary source is wastewater.** SLR models using $SAC_{254\,nm}$ perform well for 4-&5-methylbenzotriazole and DEET in catchment L (median $R^2 = 0.53$ and 0.51, median RRMSE = 51% and 57%, S3 Table 14 in S3 Tables) because they behave similarly to the indoor chemicals (S1 Fig 5 in S1 File). As with the indoor chemicals, models using $NH_4$-N perform similarly well (median RRMSE = 65% and 63%, S3 Table 17 in S3 Tables) to models using $SAC_{254\,nm}$. The addition of multiple sensor parameters does not improve the prediction significantly (Fig 3c and 3d). The models perform worse for benzotriazole in catchment L (median $R^2 = 0.23$, median RRMSE = 118%, S3 Table 14 in S3 Tables), likely because stormwater is also an important source. Model prediction based on the best sensor (Table 4) is similarly poor in the smaller catchments S and M (Fig 3c and 3d), where 4-&5-methylbenzotriazole, benzotriazole, and DEET show fluctuating concentration patterns (S1 Figs 3 and 4 in S1 File). In conclusion, linear models can only predict the concentrations of 4-&5-methylbenzotriazole, benzotriazole, and DEET when households are their primary source. However, when these chemicals originate from stormwater, they lose source-specificity, rendering predictions using the selected sensors infeasible.

### 3.4. Limitations

**General limitations of sampling strategy and data analysis.** Different numbers of samples were taken per rain event, depending on its length, and every data point is weighted equally based on our assumption that all individual measurements are equally reliable. Hence, longer rain events have a greater impact on the results than events with fewer data points. Additionally, with three to eight chemicals per chemical group, each chemical influences the group's results. For example, the individual chemicals from road runoff are strongly correlated in catchments S and L, but not in catchment M (Fig 1b), where HMMM is more rapidly mobilized at the beginning of two of the four rain events (S1 Figs 4 and 7 in S1 File). Having more rain events (> 4) and chemicals per chemical group (> 3) could improve correlation robustness. In addition, to capture the full dynamics, the beginning of the rain event needs to be sampled. The missing start of the rain event M.2 likely weakens correlations and model predictions (S1 Figs 16, 23, and 24 in S1 File). Nevertheless, M.2 was retained for analysis, as only four rain events were available in catchment M, and excluding it would have further limited the already small dataset. This suggests that sampling the beginning of a rain event is crucial, as was already recommended by Furrer et al. [32]. Another limitation is the unequal availability of sensor data across catchments. While water flow data is available everywhere, water level was only measured in catchments S and M. However, due to the strong correlation between flow and level, information on hydraulic dynamics is available for all catchments. Temperature, pH, and turbidity measurements are missing in catchments S and/or L. However, this had a minimal impact on the outcomes of this study due to the generally low correlations between these parameters and chemical concentrations. The absence of EC and $SAC_{254\,nm}$ in catchment S is more critical, as these parameters are reliable proxies for indoor chemicals in the other catchments. It is not possible to fully assess how their performance depends on catchment size. However, findings from catchments M and L suggest that EC and $SAC_{254\,nm}$ become more reliable proxies with increasing catchment size, as is the case for $NH_4$-N, which is available in all three catchments (Fig 2). In conclusion, the sampling strategy and data analysis could be improved by having a larger number of chemicals in each chemical group, by sampling more rain events, by starting the sampling shortly before the flow in the sewer increases, and by improving sensor coverage across all catchments.

**Limitations of linear regression models.** The focus of this study is on in situ water quality monitoring using online sensors. However, there may be other factors, including rainfall characteristics such as rain intensity, rain duration, and the duration of preceding dry periods [37], as well as seasonal usage patterns [32], that influence the dynamics of polar organic chemicals. Further research is needed to investigate their potential in improving predictions of organic chemical concentrations. Additionally, models are fitted to absolute concentrations of polar organic chemicals and sensor parameters. Chemical concentrations typically follow a log-normal distribution, meaning that high values disproportionately influence model performance. While log-transforming the response variable can stabilize variance and improve the model fit for lower concentrations, it shifts the model's focus away from high concentrations, which are most relevant from an ecotoxicological perspective. Therefore, we fitted concentrations on the original scale to capture absolute errors at the upper end of the concentration range. Furthermore, linear regression can occasionally predict negative concentrations, particularly when extrapolating beyond the training range, and is unable to capture non-linear relationships between organic chemicals and sensor parameters. To explore the importance of non-linear relationships, we tested random forest regression models (see S1 Fig 29 in S1 File), but found no substantial improvement in prediction accuracy. Given similar performance and the limited size of the dataset, which was insufficient to thoroughly test machine learning approaches, we selected linear regression as a robust and transparent baseline method. Finally, the linear regression models may not be universally applicable to all chemicals from the same source, as factors such as degradation or adsorption to particles can alter their behavior. Addressing these limitations in future studies with larger data sets could increase prediction accuracy for polar organic chemicals from indoor sources and road runoff.

## 3.5. Practical considerations for future application of sensor-based monitoring and remaining research needs

While our approach has certain limitations, it demonstrates strong potential for advancing the long-term monitoring of organic chemicals in CSOs, particularly for polar indoor and road runoff chemicals. In this final section, we outline key considerations for implementing sensor-based monitoring and identify remaining research needs. Specifically, we address key uncertainties regarding calibration requirements, challenges associated with sensor deployment and maintenance in urban drainage systems, as well as cost considerations.

**Prediction accuracy and calibration requirements remain to be defined.** For practical applications, it is crucial to determine the acceptable level of prediction accuracy. We consider error margins of 30% to 50% acceptable for monitoring organic chemicals in CSO discharges, given their highly dynamic conditions [13,16], as well as inherent uncertainties associated with sensor measurements and sample processing. For context, RRMSEs of 10% to 20% are typically considered satisfactory for UV-Vis measurements in urban drainage systems [50], and a 10% error is common for chemical sampling and analysis [13]. In contrast, Hubeaux et al. [27] demonstrated that micropollutant abatement in wastewater treatment plants can be predicted with errors as low as 2% based on $SAC_{254\,nm}$ measurements. However, their study relied on 24-hour composite samples and did not address the short-term fluctuations typical of sewer systems. While very high precision may be achievable in relatively stable WWTP conditions, higher error margins need to be accepted in sewer monitoring due to greater variations in the concentrations of organic chemicals. Hence, this study suggests that the accuracy of sensor-based linear regression models is satisfactory for monitoring indoor chemicals in sewer systems of larger catchments and chemicals from road runoff. However, whether the model accuracy is satisfactory ultimately depends on the requirements of the intended application.

The sensor-based monitoring approach developed in this study requires local calibration of the regression models. At each monitoring site, sufficient reference samples must be collected and analyzed to achieve satisfactory model performance. The necessity of local calibration was also highlighted in studies using UV absorbance as feedback control parameters for micropollutant abatement in WWTPs, due to differences in wastewater composition [29]. Our study suggests that a minimum of 4 to 13 rain events, containing an average of 18 samples, may be required for reliable model calibration. Overfitting was an issue for models in catchment M, which used data from only four events, whereas catchment L demonstrated good model performance with data from 13 events. However, more specific calibration requirements remain

unclear and likely depend on catchment size and the variability of the substances and rain events monitored [19]. Mutzner et al. [2] reported a median minimum number of seven events for estimating event mean concentrations of organic chemicals in CSOs. Further studies should assess the minimum number of events needed for reliable calibration and investigate how this requirement varies with catchment size, characteristics, and selected chemicals. A practical approach is to apply an iterative cross-validation procedure, where rain events are sampled until the prediction error falls below a predefined threshold [51,52]. In summary, while our modelling approach shows promise, its future application by utilities remains limited due to unclear calibration demands.

**Sensor-based monitoring requires careful planning and sensor maintenance.** Sensor-based monitoring is only effective if the sensors provide reliable, high-quality data. However, the environmental conditions in urban drainage systems are harsh, including variable flow regimes, corrosive gases, high humidity, and solid loads, which can impair sensor performance and data quality [22,53,54]. Frequent maintenance, including sensor cleaning and recalibration, is therefore essential to prevent sensor drift and biofilm formation, thereby ensuring the long-term stability of data and model performance [33]. The quality of the data used in this study was ensured through regular sensor maintenance (see S1 Table 7 in S1 File), which requires trained staff. For example, flow sensors were cleaned one to four times per year, while UV-Vis probes were cleaned on a weekly or biweekly basis (S1 Table 7 in S1 File). Where feasible, automated sensor cleaning with compressed air was used to further enhance sensor performance [33]. Lechevallier et al. [33] also demonstrated the value of automated quality control, including outlier detection, drift analysis, and weekly data inspection. However, further research is needed to evaluate the impact of reduced maintenance on sensor performance and model stability over longer periods. Additionally, access to underground urban drainage structures is often difficult, making it challenging to install and maintain sensors. For example, Furrer et al. [13] selected monitoring sites at the inlets of WWTPs in catchments S and L based on their good accessibility, which facilitated sensor installation and maintenance. In summary, regular maintenance, accessible monitoring sites, and trained personnel are essential for the successful implementation of the sensor-based monitoring approach developed in this study.

**Sensor-based monitoring is a cost-effective approach for real-time monitoring of organic chemicals.** As previously discussed, calibrating the sensor-based organic chemical prediction models requires chemical analysis of 4 to 13 rain events, each with an average of 18 samples. The calibration phase is relatively costly as it involves both sensor measurements and chemical analysis (see S1 Tables 8 and 9 in S1 File). However, once the model is calibrated, the return on investment is rapid due to the high per-sample cost of laboratory-based LC-MS analysis (S1 Table 9 in S1 File). Based on the cost estimates in S1 Tables 8 and 9 in S1 File, the investment costs in sensors can be offset after monitoring a single rain event, even when using more expensive sensors such as UV-Vis probes or ion-selective electrodes. Maintenance costs are expected to be similar to or lower than those of the traditional autosampler-based approach, depending on the number and type of sensors. The traditional monitoring approach also involves recurring costs, as site visits are necessary for sample collection after every rain event. Beyond cost savings, sensor-based monitoring offers additional advantages, including high temporal resolution, continuous measurements, and real-time data availability, features that are particularly valuable in highly dynamic environments, such as urban drainage systems. In summary, the sensor-based approach enables substantial cost reduction for long-term experimental campaigns while maintaining satisfactory accuracy for indoor and road runoff chemicals.

## 4. Conclusions

The assessment of online sensors as proxies for polar organic chemicals in three catchments of different sizes leads to the following conclusions:

- Indoor chemicals show a source-specific dilution pattern during wet-weather events. This pattern is more stable in larger catchments (>20'000 inhabitants), whereas indoor chemicals exhibit more pronounced fluctuations due to pulse emissions in smaller catchments. Chemicals from road runoff show source-specific dynamics independent of catchment size.

- Level, flow, EC, SAC$_{254 nm}$, and NH$_4$-N are suitable sensor proxies for polar indoor chemicals in larger catchments (M and L), where these chemicals show a pronounced dilution pattern. Linear regression models using SAC$_{254 nm}$ and NH$_4$-N perform best, with median prediction errors of 32% and 29%, respectively, in the large catchment (approx. 200,000 PE). The model's prediction performance is worse for indoor chemicals in smaller catchments, likely due to stronger concentration fluctuations.

- No universally valid sensor proxy has been identified for polar organic chemicals from road runoff. Water flow and level have been identified as the most effective proxies, yielding median prediction errors of 39 to 44%. However, the role of environmental factors that potentially affect chemical leaching, such as antecedent dry period, rain intensity, or traffic volume, is currently unclear or limited by data availability [32,47,48,55]. Further research is needed to identify environmental factors or sensor proxies that better reflect the complex release mechanisms of these chemicals, given their ecotoxicological relevance [56,57].

- No suitable sensor proxies could be identified for PPPs, biocides, and chemicals originating from many diverse sources due to their non-source-specific nature and the lack of sensor proxies reflecting their highly variable concentrations. Given the potential toxicity of these substances, their monitoring is critical, but must continue to rely on traditional monitoring methods for the foreseeable future.

- Adding more sensors as predictors generally does not improve prediction performance significantly (change in median prediction error < 10%), because they often capture noise rather than actual chemical patterns. Thus, online measurements with a single sensor parameter (level, flow, SAC$_{254 nm}$, NH$_4$-N) are promising for the monitoring of indoor and road runoff chemicals.

- The regression models developed in this study require site-specific calibration based on sample collection and chemical analysis. Our analysis suggests that at least 4 to 13 events may be necessary for model calibration, depending on catchment characteristics and chemical variability.

- Sensor-based monitoring has the potential for real-world implementation and for informing mitigation strategies to reduce the environmental impact of CSO discharges. However, further refinements are necessary. Future research should focus on defining calibration requirements, expanding datasets with additional catchments, chemicals, and influence factors, and improving model accuracy. Hybrid approaches that combine sensor monitoring with traditional sampling and LC-MS analysis for selected samples could be explored as they may increase the robustness of chemical predictions.

## Supporting information

**S1 File. Supporting information.** This file provides additional information on statistical analyses, methodological details, and supporting data visualization.
(PDF)

**S2 Tables. Correlation tables.** The tables in this zip file show the statistical measures (Pearson correlation coefficient, p-value, confidence interval, etc.) of all correlations calculated for this manuscript. A ReadMe file explains further details about the content and structure of each table.
(ZIP)

**S3 Tables. Model tables.** The tables in this zip file show the statistical measures ($R^2$, RMSE, *p*-value, confidence intervals, etc.) of all models calculated for this manuscript and the predicted chemical concentrations. A ReadMe file explains further details about the content and structure of each table.
(ZIP)

## Acknowledgments

We thank Andreas Scheidegger for his valuable support with the statistical models. We are also very grateful to Simon Bloem and Michael Arnold for sharing their detailed knowledge on sensor installation and maintenance.

## Author contributions

**Conceptualization:** Laura Waldner, Viviane Furrer, Pierre Lechevallier, Heinz Singer, Lena Mutzner.

**Data curation:** Laura Waldner, Viviane Furrer, Pierre Lechevallier.

**Formal analysis:** Laura Waldner.

**Funding acquisition:** Lena Mutzner.

**Investigation:** Laura Waldner, Viviane Furrer, Pierre Lechevallier, Fabienne Maire.

**Methodology:** Laura Waldner, Viviane Furrer, Pierre Lechevallier, Fabienne Maire, Heinz Singer, Lena Mutzner.

**Project administration:** Laura Waldner, Lena Mutzner.

**Resources:** Viviane Furrer, Pierre Lechevallier, Heinz Singer, Lena Mutzner.

**Software:** Laura Waldner.

**Supervision:** Lena Mutzner.

**Validation:** Laura Waldner, Viviane Furrer, Pierre Lechevallier.

**Visualization:** Laura Waldner.

**Writing – original draft:** Laura Waldner.

**Writing – review & editing:** Laura Waldner, Viviane Furrer, Pierre Lechevallier, Fabienne Maire, Heinz Singer, Lena Mutzner.

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
