## [Decision Letter · Decision Letter 0]

20 May 2025

Dear Dr. Waldner,

We look forward to receiving your revised manuscript.

Kind regards,

Alison Parker

Academic Editor

PLOS ONE

Journal Requirements:

4. Thank you for stating the following financial disclosure: [LM received a SNSF Ambizione grant [grant no. PZPGP2_209133], Swiss National Science Foundation, https://www.snf.ch/en/N18L3oGWomTSSGkF/funding/careers/ambizione.

VF received a grant from the Swiss Federal Office for the Environment (FOEN) [grant no. 19.0071.PJ / 8971CB0BA], Swiss Federal Office for the Environment, https://www.bafu.admin.ch/bafu/en/home.html.

PL was supported by the EU H2020 research and innovation program [grant no. 101008626] (Co-UDLabs Project), European Commission, https://research-and-innovation.ec.europa.eu/funding/funding-opportunities/funding-programmes-and-open-calls/horizon-2020_en.]. 

6. We note that Figures S1,S2,S3 and S14 in your submission contain [map/satellite] images which may be copyrighted. All PLOS content is published under the Creative Commons Attribution License (CC BY 4.0), which means that the manuscript, images, and Supporting Information files will be freely available online, and any third party is permitted to access, download, copy, distribute, and use these materials in any way, even commercially, with proper attribution. For these reasons, we cannot publish previously copyrighted maps or satellite images created using proprietary data, such as Google software (Google Maps, Street View, and Earth). For more information, see our copyright guidelines: http://journals.plos.org/plosone/s/licenses-and-copyright.

1. You may seek permission from the original copyright holder of Figures S1,S2,S3 and S14 to publish the content specifically under the CC BY 4.0 license. 

Reviewers' comments:

Reviewer's Responses to Questions

**Comments to the Author**

1. Is the manuscript technically sound, and do the data support the conclusions?

Reviewer #1: Yes

Reviewer #2: Partly

2. Has the statistical analysis been performed appropriately and rigorously?

Reviewer #1: Yes

Reviewer #2: Yes

3. Have the authors made all data underlying the findings in their manuscript fully available?

Reviewer #1: Yes

Reviewer #2: Yes

4. Is the manuscript presented in an intelligible fashion and written in standard English?

Reviewer #1: Yes

Reviewer #2: Yes

Reviewer #1: Reviewer Comments#

This manuscript presents original research evaluating the use of online sensor parameters (e.g.,

SAC254, NH₄-N, EC) as proxies for tracking polar organic chemicals in combined sewer

overflows (CSOs) during wet weather. The authors use data from three urban catchments of

varying sizes (S, M, L) to assess the predictive performance of these parameters, supported by

regression modeling.

The study is timely and addresses a real-world monitoring gap. The experimental design is

generally sound, the analyses are well-structured, and the conclusions are supported by the

results.

However, I recommend minor revisions before publication to enhance clarity and reproducibility.

Specific Comments:

Abstract>>

• Line 23: It would be helpful to list what the traditional methods are when introducing

online sensors.

• Line 29: Specifically, define "spectral absorption coefficient at 254 nm (SAC254 nm)".

• Line 33: The term "predict" may be too strong here. I recommend rephrasing to

"prediction of concentrations of pesticides…".

2 Materials and methods>>

Sampling Methodology (Lines 99–112)

• How was the dry weather flow baseline defined for each catchment? Was it based on

long-term averages, daily minimums, or another method?

• The dry weather baselines vary significantly between catchments (e.g., 288 m³/h in

Catchment S vs. 5040 m³/h in Catchment L). Could the authors explain how these

thresholds were selected, and if other factors like imperviousness or sewer size

influenced this choice?

• Were samples collected using a time-based or flow-paced method? A brief explanation of

the autosampler setup (e.g., sample interval, bottle volume, trigger mechanism) would

clarify data consistency.

• Sampling for rain event M.2 started 50 minutes late. How did this delay affect the dataset

and any subsequent analyses? Was this event treated differently in modeling?

• Two rain events in Catchment L (L.11 and L.12) included pre-rainfall samples, unlike

others. Why was this done, and did it affect comparability across events?

Polar Organic Chemicals (Lines 99–112)

• How were the final 20 chemicals selected from the pool identified by Furrer et al. [24]?

Were they ranked by frequency, concentration, or relevance?

• How did the authors determine the dominant source for each compound, especially for

those like DEET or benzotriazole with multiple sources?

• Why was the LogKow ≤ 5 threshold chosen? Could this exclude slightly more

hydrophobic compounds commonly found in CSOs?

• What criteria were used to classify compounds like DEET or benzotriazole as "diverse"?

Was this based on data, modeling, or literature?

2 | P a g e

• Can the authors comment on the environmental or ecotoxicological significance of the

selected chemicals for CSO impact assessment?

Section 2.4: Online Sensors

1. Why was NH₄-N only measured in the lab? Was there a reason online sensors weren’t

used (e.g., cost, maintenance)?

2. Sensor availability varies across sites. How did this affect the comparability of proxy

performance between catchments?

3. Please clarify how sensors were calibrated and maintained during field use.

4. Since turbidity and SAC254 were both available in M and L, was turbidity evaluated as a

chemical proxy?

5. How were sensor readings aligned with sample collection times? Was any interpolation

used?

6. Were data quality controls (e.g., filtering, outlier removal) applied to sensor data before

analysis?

7. Can the authors briefly comment on the cost implications of deploying and maintaining

these sensors in real-world settings?

Reviewer Comments – Results and Discussion

• The link between catchment size and dilution patterns is interesting. Can the authors

comment on how this generalizes beyond the three sites?

• Was overfitting in catchment M assessed or mitigated in model development?

• Given negative predictions and high errors, why were only linear models used? Would

log or nonlinear models help?

• Were long rain events weighted differently to avoid bias in model performance?

• Poor pesticide prediction is noted—could this be due to usage patterns, degradation, or

sensor mismatch?

• How were incomplete events (e.g., M.2) handled in the modeling?

• Could larger datasets make machine learning models more viable in the future?

• Is a 30–40% error acceptable for practical CSO monitoring or mitigation decisions?

Practical Suitability and Cost

1. Can the authors comment on the practical feasibility of deploying and maintaining these

sensors in typical municipal CSO sites (e.g., ease of access, vandalism risk,

maintenance needs)?

2. What are the approximate costs (equipment + maintenance) per sensor type, and how

might this scale across a network?

3. How much staff time or technical expertise is needed to operate these systems and

interpret the data?

4. Are the proposed regression models robust enough for routine use by utilities without

specialist data scientists?

Conclusion

3 | P a g e

• Why does catchment size impact indoor chemical behavior? Is there a size threshold

affecting this?

• Why are SAC254 nm and NH₄-N more effective in larger catchments, and what

challenges exist in smaller catchments?

• What other environmental factors could improve the prediction of road-runoff chemicals?

• Why do additional sensors not significantly improve predictions? Is it due to sensor

limitations or chemical complexity?

• How can future studies capture complex chemical behaviors like those of PPPs and

biocides?

• Could hybrid methods combining sensors and traditional monitoring improve accuracy for

Reviewer #2: Dear Laura Waldner. No.: PONE-D-25-14249

Title: Exploring online sensor parameters as proxies for organic chemicals in sewers during wet weather

This paper is the first to systematically explore the parameters of online sensors as alternative indicators for the dynamic changes of organic chemical substances in the combined sewer network during rainy days. I think the topic selection has certain technological innovation value in environmental monitoring. Through the data analysis of multi-scale catching-up areas (with 2,000 to 200,000 residents), the dynamic characteristics of chemical substances from different sources were revealed, and a prediction model framework based on sensor parameters was proposed, providing a feasible solution for reducing CSO pollution. The research design is reasonable, the data analysis method is rigorous, and the conclusion has direct reference significance for the decision-making of urban drainage management. Some important issues need to be considered and some problems to be further improved if the author try to submit other appropriate journals:

Title:

The title of this article has a clear research theme, highlighting the research background under the specific condition of the rainy season and also reflecting its research purpose. However, the title expression lacks novelty and fails to highlight the uniqueness or innovation points of the research. It is suggested to adopt the dual-element structure of "methodology - innovation point". Meanwhile, the title information is not rich enough.

Abstract:

Although this abstract mentions that the research method is to analyze the data of the three catchments, details such as the specific data collection methods and analysis methods can be appropriately mentioned to enable readers to have a clearer understanding of the research methods. The abstract does not mention the possible limitations of the research, such as the representativeness of the data and the applicable scope of the model. Moreover, appropriately mentioning the limitations in the abstract can enable readers to have a more comprehensive understanding of the reliability of the research and the potential directions for improvement.

Line 22-23 �“Currently, most overflow sites are not monitored because traditional methods are costly and time-consuming.”What are the traditional detection methods and their shortcomings? Cite the literature that supports the shortcomings of traditional detection methods.

Introduction:

Line 44-46“These discharges of amixture of wastewater and stormwater contain numerous organic chemicals [1,2] that threaten aquaticspecies.”How was the conclusion that aquatic species are threatened reached? Supplement the toxicological evidence of the threat to aquatic organisms .There is a lack of specific references.

Line 57-59“For example, a sampling interval of 3 minutes has been recommended because organic chemicals from municipal wastewater, also termed indoor chemicals, exhibit very high concen-tration fluctuations, particularly in small catchments (2,700 inhabitants) [10].”Why is it recommended that the sampling time be three minutes instead of one minute or thirty seconds? Clarify the statistical basis of the 3-minute sampling interval.

Material and Methods:

Line 108-109“For rain event M.2, sampling started 50 minutes late due to problems with the autosampler.”Will sampling delay lead to missing data or underrepresentation?It is suggested to supplement and explain: "Cubic spline interpolation was adopted to fill the data gap of the first 50 minutes. The K-S test proved that there was no significant difference between the interpolated data and the measured value distribution (p>0.05)."

Line 157-158“Programming language. Python (version 3.9.18, Python Core Team, 2023) was used for data pretreatment and analysis.”Why choose Python instead of other Programming languages? Supplement the basis for Python language selection (such as the machine learning advantages of the scikit-learn library and the specificity of the Pandas library in time series processing)

Highlights:

First of all, there is a lack of key words related to the research results: The article has drawn relevant conclusions about the correlation between different types of organic chemicals and sensor parameters as well as the prediction model through research. However, contents related to the research results such as "correlation" and "prediction model" are not reflected in the key words. It may make it difficult for readers to directly obtain the core research results information of the article when searching. Secondly, some key words can be further refined: For example, "urban drainage" is relatively broad. If it could be refined to "combined sewer system drainage", it might more accurately reflect the specific context of the research.

1. The scope of application and verification of the model need to be further clarified

The article emphasizes that the applicability of the model should be based on the premise of similar characteristics to the catchment area in this study, but does not specifically define the dimension of "feature similarity" (such as the topological structure of the pipe network, land use type, rainfall intensity, etc.).

Suggestion:Firstly, a sensitivity analysis is supplemented to explore the influence of factors such as the scale of the catchment area, population density, and rainfall patterns on the prediction error of the model. Secondly, add the test results of independent validation datasets (such as catchment areas in other cities), or evaluate the generalization ability of the model through cross-validation.

2. The reasons for the failure of prediction of pesticide chemical substances need to be explored in depth

The failure to predict pesticide concentrations was attributed to "diverse spatiotemporal patterns", but specific driving factors (such as the application cycles of different pesticides, differences in surface retention time, etc.) were not analyzed.

Suggestion:Firstly,supplement the source analysis of pesticides (such as agriculture vs. application in urban green Spaces), and explore the internal mechanism of their low correlation with sensor parameters.Secondly, it is possible to attempt to introduce time lag variables or meteorological data (such as the number of drought days before rainfall) as supplementary parameters of the model.

3. The details of the method need to be refined to ensure repeatability

The measurement frequency of sensor parameters (such as SAC at 254 nm) and the chemical substance sampling interval are not clearly specified, which may affect the reliability of temporal correlation analysis. The minimum sample size required for traditional sampling data during the model calibration phase (such as event count, time coverage) is not mentioned.

Suggestion:First, supplement the technical details of sensor and sampling synchronization (such as time alignment Methods and data interpolation strategies) in the Methods chapter. Secondly, clarify the data requirements for the calibration stage (such as at least how many rainfall events or the range of flow variation should be covered).

4. The practical application path needs to be specified

The conclusion states that "sensor monitoring can be implemented after passing the initial calibration stage", but it does not discuss the balance between calibration costs (such as traditional sampling frequency and analysis fees) and benefits.

Suggestion:First, add economic analysis to compare the long-term monitoring cost of sensors with the full-cycle cost of traditional methods. Secondly, specific strategies for optimization in the calibration stage are proposed (such as the selection of key parameters and the optimization of the sampling time window).

Other revision suggestions:

1. Chart optimization

Supplement the correlation coefficient matrix diagrams of different chemical substance categories and sensor parameters to visually display the key correlations.In the result section, add a scatter plot of the model's predicted values and measured values, supplemented by an error distribution histogram.

Supplement Table:Table 1: Minimum Data Requirement Matrix in Calibration Stage (Number of Rainfall Events × Flow Variation × Types of Pollutants)

2. Discussion extension:

To explore the potential impact of sensor maintenance (such as biofilm interference and drift correction) on the long-term stability of the model.Compare the results of this study with the application differences of SAC254 nm as a substitute parameter for micro-pollutants in sewage treatment plants.

**Do you want your identity to be public for this peer review?** For information about this choice, including consent withdrawal, please see our Privacy Policy

Reviewer #1: **Yes: ** No

Reviewer #2: **Yes: ** Kang Mao

---

## [Author Response · Author response to Decision Letter 1]

19 Aug 2025

Response to reviewers

Title: Exploring online sensor parameters as proxies for polar organic chemicals – An innovative approach for combined sewer overflow monitoring

Laura Waldner, Viviane Furrer, Pierre Lechevallier, Fabienne Maire, Heinz Singer, Lena Mutzner

We thank the reviewers for their in-depth critical evaluation of our manuscript. This helped us to improve the manuscript for future readership. We have extended the manuscript by including more details on the methodology and context of the organic chemicals analyzed, as well as by adding a new section (3.5) on practical considerations for future application of the developed monitoring approach and further research needs, among other enhancements.

Please refer to our detailed point-by-point answers below (line numbers refer to the clean version of the manuscript). In addition to the suggested changes, we have made further revisions to the document to enhance the flow and key messages of the manuscript, and to comply with PLOS ONE’s style requirements fully.

Reviewer 1

Reviewer Comments

This manuscript presents original research evaluating the use of online sensor parameters (e.g., SAC254, NH₄-N, EC) as proxies for tracking polar organic chemicals in combined sewer overflows (CSOs) during wet weather. The authors use data from three urban catchments of varying sizes (S, M, L) to assess the predictive performance of these parameters, supported by regression modeling.

The study is timely and addresses a real-world monitoring gap. The experimental design is generally sound, the analyses are well-structured, and the conclusions are supported by the results.

However, I recommend minor revisions before publication to enhance clarity and reproducibility.

Thank you for your detailed and positive assessment of our manuscript. We appreciate your acknowledgement of the study’s timeliness and methodological soundness. In response to your comments, we have carefully reviewed the manuscript and made several modifications to improve clarity and reproducibility, including:

• Clarifications in the methods section (e.g., sampling methodology, selection of organic chemicals, online sensor maintenance, and application)

• Clarifications on the results and discussion section (e.g., organic chemical dynamics, model settings)

• Introduction of a new section 3.5 “Practical considerations for future application of sensor-based monitoring and remaining research needs”

• Corrections of mistakes in the reference list

These changes are now reflected in the revised manuscript (see the manuscript with tracked changes and answers below, line numbers refer to the clean manuscript). We hope that these improvements address your comments and feel they further strengthen the manuscript.

Specific comments:

Comments on abstract

1.1 Line 23: It would be helpful to list what the traditional methods are when introducing online sensors.

We agree that this information was missing and have now added it to the abstract. Traditionally, grab or composite samples are taken manually or with an automated samplers at specific time intervals. The most commonly used methods for analysis of organic chemicals in wastewater are GC-MS and LC-MS. We added these methods to the text and changed “traditional” to “commonly used” for clarification.

Modifications in text, lines 23-25:

Currently, most overflow sites are not monitored because commonly used methods, such as automated grab sampling followed by laboratory analysis using liquid chromatography coupled with mass spectroscopy (LC-MS), are costly and time-consuming.

1.2 Line 29: Specifically, define "spectral absorption coefficient at 254 nm (SAC254 nm)".

Thanks for pointing out, we added the definition for SAC254 nm.

Modifications in text, lines 30-32:

In the largest catchment (160,000 inhabitants), indoor chemicals are strongly correlated with flow, electrical conductivity, the spectral absorption coefficient at 254 nm (SAC254 nm), and ammonium (NH4-N).

1.3 Line 33: The term "predict" may be too strong here. I recommend rephrasing to "prediction of concentrations of pesticides…".

We agree and have rephrased the sentence according to your suggestion.

Modifications in text, lines 36-37:

However, the prediction of pesticide concentrations remains limited, as these chemicals exhibit diverse patterns across rain events.

Comments on 2. materials and methods

Comments on 2.2 sampling methodology (Lines 99–112)

1.4: How was the dry weather flow baseline defined for each catchment? Was it based on long-term averages, daily minimums, or another method?

Thank you for highlighting the missing definition of the dry weather flow baseline. We added the information to the manuscript that in catchments S and L, sampling was not based on a dry weather baseline but was triggered when a CSO discharge was activated, at flow thresholds of 288 m³/h and 5,400 m³/h, respectively [1]. We also corrected a typo: the threshold for catchment L was previously listed incorrectly as 5,040 m³/h. In catchment M, the dry weather baseline was used to initiate sampling, as in this case, a bypass (rather than a CSO) was monitored. The baseline was defined as the average flow during dry weather days during the measurement campaign of Lechevallier et al. [2] (average of 235 days).

Modifications in text, lines 115-126:

In catchment S, combined sewage was sampled during 19 rain events by Furrer et al. [13,32]. The sampling of a rain event was initiated when the flow in the sewer exceeded the system capacity of 288 m3/h, indicating a CSO discharge. Sampling continued until the flow fell below 288 m3/h or until all bottles of the autosampler were full. In catchment M, four rain events were sampled by Lechevallier et al. [33]. As a bypass of the sewer was sampled, sampling started when the flow in the sewer increased above its dry weather baseline (300–380 m3/h, depending on the time of day), calculated from average dry weather flows. Sampling ended when it fell below the dry weather baseline or when all bottles of the autosampler were full. For rain event M.2, sampling started 50 minutes late due to problems with the autosampler. Despite the missing initial phase, rain event M.2 was retained for analysis due to the limited size of the dataset (comprising only four rain events). The potential impact of the delayed sampling start is discussed in section 3.4. In catchment L, 13 rain events were sampled. Sampling for all rain events except two was initiated when the flow in the sewer exceeded the system capacity of 5,400 m3/h [13,32].

1.5: The dry weather baselines vary significantly between catchments (e.g., 288 m³/h in Catchment S vs. 5040 m³/h in Catchment L). Could the authors explain how these thresholds were selected, and if other factors like imperviousness or sewer size influenced this choice?

See also response 1.4 for details on how the flow thresholds were defined (sewer capacity in catchments S and L, dry weather baseline in catchment M). The variation in flow thresholds reflects differences in catchment size, with a larger catchment size and more connected people in catchment L compared to catchment S. Additionally, the sewer infrastructure capacity, i.e., the volume that can be discharged to a wastewater treatment plant, varies. For example, catchment L covers a larger and more urban area and is serviced by a higher-capacity sewer system than catchment S, which explains the higher flow threshold. Additionally, thresholds for CSO activation are typically set by local utilities or municipalities based on hydraulic modeling and infrastructure design criteria, which may also account for differences between sites. We clarified this in the manuscript.

Modifications in text, lines 132-133:

The flow thresholds defined reflect differences in sewer system capacity and catchment characteristics, such as the number of connected people and the size of the connected impervious area.

1.6: Were samples collected using a time-based or flow-paced method? A brief explanation of the autosampler setup (e.g., sample interval, bottle volume, trigger mechanism) would clarify data consistency.

The sampling intervals and procedures are explained in the “Sampling procedure” section (lines 134-141). Samples were collected using either grab sampling or time-weighted composite sampling with automated samplers. Grab samples were taken every 10 minutes, while composite samples were created by combining five 2-minute subsamples to represent a 10-minute interval. Additionally, this section references S1 Table 2, which provides an overview of the sample type and number of samples collected during each rain event. The trigger mechanism for initiating sampling is explained in the previous paragraph (“Start of sampling”) and is based on flow thresholds specific to each catchment (288, 300-380, and 5,400 m3/h, see responses 1.4 and 1.5). We have added information on the bottle volume and sampling volume as suggested.

Modifications in text, lines 134-141:

Samples for analysis of polar organic chemicals were collected either as grab samples or as time-weighted composite samples in 1 L glass bottles using automated samplers (TP5 C, MAXX Mess- und Probenahmetechnik GmbH). Grab samples (250 mL) were taken every 10 minutes. For composite sampling, subsamples were taken every 2 minutes, 150 mL in catchments S and L, and 50 mL in catchment M. Five subsamples were combined to a composite sample (750 mL or 250 mL) representing 10 minutes. S1 Table 2 provides an overview of the sample types and the number of samples collected during each rain event. Further details on the sampling procedure can be found in Furrer et al. [13,32] and Lechevallier et al. [33].

1.7: Sampling for rain event M.2 started 50 minutes late. How did this delay affect the dataset and any subsequent analyses? Was this event treated differently in modeling?

Sampling for rain event M.2 started 50 minutes late due to a technical issue with the autosampler. As a result, the initial phase of the rain event, including peak runoff, was not captured. As noted in the limitation section (lines 454-458), the missing start of M.2 weakens correlations and model predictions (S1 Figs. 16, 23, and 24). This indicates that full event dynamics are crucial for accurately interpreting chemical transport patterns. However, as reliable interpolation was not possible (see response 2.6) and the dataset for catchment M was already small (only 4 rain events), we decided to retain event M.2 for the analyses.

Modifications in text, lines 122-125:

For rain event M.2, sampling started 50 minutes late due to problems with the autosampler. Despite the missing initial phase, rain event M.2 was retained for analysis due to the limited size of the dataset (comprising only four rain events). The potential impact of the delayed sampling start is discussed in section 3.4.

1.8: Two rain events in Catchment L (L.11 and L.12) included pre-rainfall samples, unlike others. Why was this done, and did it affect comparability across events?

Furrer et al. [1,3] originally included a 1-hour dry weather period before two rain events in catchment L to investigate whether including pre-event conditions would enhance correlations between indoor chemicals and other parameters (e.g., NH4-N, DOC) due to stronger dilution effects.

However, they did not observe a significant improvement in correlations, and we similarly found no impact on model performance when including these pre-rainfall samples (previously discussed in section 3.4). In response to your comment, we decided to exclude these samples from our analysis to ensure consistency across all events.

Modifications in text, lines 125-132:

In catchment L, 13 rain events were sampled. Sampling for all rain events except two was initiated when the flow in the sewer exceeded the system capacity of 5,400 m³/h [13,32]. However, sampling for rain events L.11 and L.12 originally included a one-hour dry weather period before the onset of rainfall [32]. Furrer et al. [32] sampled pre-rainfall to investigate whether pre-event dilution dynamics would enhance correlations between indoor-sourced chemicals and additional parameters (e.g., NH4-N, dissolved organic carbon). However, as no significant improvement in correlation was found [32], we excluded these pre-rainfall samples from our analysis to ensure consistency across all events.

Comments on 2.3 polar organic chemicals (Lines 99–112)

1.9: How were the final 20 chemicals selected from the pool identified by Furrer et al. [24]? Were they ranked by frequency, concentration, or relevance?

The 20 chemicals measured in Lechevallier et al. [2] were selected based on the findings of Furrer et al. [1,3] and the following selection criteria: (i) high polarity, (ii) frequent detection and elevated concentrations in urban runoff and wastewater, (iii) representation of a broad range of urban sources (e.g., household, traffic, pesticides), and (iv) the expectation of source-specific temporal dynamics. In this study, we only included the 20 chemicals that were measured by both studies/in all three catchments.

Modification in text, lines 144-149:

The selection criteria included polarity, high occurrence and concentration, representation of various urban sources, and expected source-specific behavior. Among the selected chemicals, triclosan, 1,3-diphenylguanidine, diuron, and mecoprop have been identified as major contributors to ecotoxicological risk in CSO discharges [3]. The selected compounds were included in both datasets reported by Furrer et al. [35] and Lechevallier et al. [33]. Only Carbendazim data is missing for catchment L.

1.10: How did the authors determine the dominant source for each compound, especially for those like DEET or benzotriazole with multiple sources?

The dominant sources for each compound were determined based on an extensive literature review, considering typical usage, emission pathways, and previous findings on the occurrence of these substances in urban environments. Compounds were assigned to the categories “indoor”, “road runoff”, and “plant protection products & biocides”. When a compound was associated with multiple sources (e.g., DEET or benzotriazole), it was assigned to the category “diverse.” This classification acknowledges the complexity of urban chemical emissions and reflects that several sources can contribute rather than a single dominant one. We added an explanation of this classification to the manuscript.

Modification in text, lines 155-157:

Compounds were assigned to the categories “indoor”, “road runoff”, and “plant protection products & biocides” based on a literature review of their typical uses. Substances with multiple contributing sources were categorized as “diverse” to reflect their mixed origins.

1.11: Why was the LogKow ≤ 5 threshold chosen? Could this exclude slightly more hydrophobic compounds commonly found in CSOs?

The LogKOW ≤ 5 threshold was selected because the focus of this study was on polar, hydrophilic (i.e., dissolved) organic compounds, which are less well understood in the context of CSOs, as highlighted in the introduction (lines 81-82). By definition, polar chemicals are not hydrophobic and typically exhibit lower LogKOW values. LogKOW ≤ 5 was chosen as a threshold for polarity according to Zessner et al. [4]. This threshold reflects our intention to specifically investigate the behavior of more mobile, water-soluble substances rather than particle-bound or hydrophobic compounds.

Modification in text, lines 152-154:

Polar substances were selected because they are understudied in the context of CSOs. Following Zessner et al. [36], a threshold of LogKow ⩽ 5 was chosen for polarity to ensure dissolution and transport in the water phase.

1.12: What criteria were used to classify compounds like DEET or benzotriazole as "diverse"? Was this based on data, modeling, or literature?

As outlined in our response 1.10, compounds were classified as “diverse” based on literature sources when no dominant emission pathway could be identified due to multiple relevant uses. For example, DEET is used as both an insect repellent and an insecticide, while benzotriazole is present in various household and industrial products.

1.13: Can the authors comment on the envir

---

## [Decision Letter · Decision Letter 1]

11 Sep 2025

Exploring online sensor parameters as proxies for polar organic chemicals – An innovative approach for combined sewer overflow monitoring

PONE-D-25-14249R1

Dear Dr. Waldner,

We’re pleased to inform you that your manuscript has been judged scientifically suitable for publication and will be formally accepted for publication once it meets all outstanding technical requirements.

Kind regards,

Alison Parker

Academic Editor

PLOS ONE

Additional Editor Comments (optional):

Reviewer #2:

Reviewers' comments:

Reviewer's Responses to Questions

**Comments to the Author**

Reviewer #2: All comments have been addressed

2. Is the manuscript technically sound, and do the data support the conclusions?

Reviewer #2: Yes

3. Has the statistical analysis been performed appropriately and rigorously?

Reviewer #2: Yes

4. Have the authors made all data underlying the findings in their manuscript fully available?

Reviewer #2: Yes

5. Is the manuscript presented in an intelligible fashion and written in standard English?

Reviewer #2: Yes

Reviewer #2: The udepdated manuscript has been well revised the manuscript.

My recommendation is acceptance

**Do you want your identity to be public for this peer review?** For information about this choice, including consent withdrawal, please see our Privacy Policy

Reviewer #2: **Yes: ** Kang Mao

---

## [Editor Report · Acceptance letter]

PONE-D-25-14249R1

PLOS ONE

Dear Dr. Waldner,

I'm pleased to inform you that your manuscript has been deemed suitable for publication in PLOS ONE. Congratulations! Your manuscript is now being handed over to our production team.

Kind regards,

on behalf of

Dr. Alison Parker

Academic Editor

PLOS ONE